# Silibinin and SARS-CoV-2: Dual Targeting of Host Cytokine Storm and Virus Replication Machinery for Clinical Management of COVID-19 Patients

**DOI:** 10.3390/jcm9061770

**Published:** 2020-06-07

**Authors:** Joaquim Bosch-Barrera, Begoña Martin-Castillo, Maria Buxó, Joan Brunet, José Antonio Encinar, Javier A. Menendez

**Affiliations:** 1Medical Oncology, Catalan Institute of Oncology (ICO), Dr. Josep Trueta Hospital of Girona, 17007 Girona, Spain; jbrunet@iconcologia.net; 2Department of Medical Sciences, Medical School University of Girona, 17003 Girona, Spain; 3Unit of Clinical Research, Catalan Institute of Oncology, 17007 Girona, Spain; bmartin@iconcologia.net; 4Girona Biomedical Research Institute, 17190 Salt, Girona, Spain; mbuxo@idibgi.org; 5Catalan Institute of Oncology, IDIBELL, 08908 L’Hospitalet de Llobregat, Barcelona, Spain; 6Institute of Research, Development and Innovation in Biotechnology of Elche (IDiBE) and Molecular and Cell Biology Institute (IBMC), Miguel Hernández University (UMH), 03202 Elche, Spain; 7Program Against Cancer Therapeutic Resistance (ProCURE), Metabolism and Cancer Group, Catalan Institute of Oncology, 17007 Girona, Spain

**Keywords:** coronavirus, stat3, cytokine storm, IL-6, JAK, remdesivir, senescence

## Abstract

COVID-19, the illness caused by infection with the novel coronavirus SARS-CoV-2, is a rapidly spreading global pandemic in urgent need of effective treatments. Here we present a comprehensive examination of the host- and virus-targeted functions of the flavonolignan silibinin, a potential drug candidate against COVID-19/SARS-CoV-2. As a direct inhibitor of STAT3—a master checkpoint regulator of inflammatory cytokine signaling and immune response—silibinin might be expected to phenotypically integrate the mechanisms of action of IL-6-targeted monoclonal antibodies and pan-JAK1/2 inhibitors to limit the cytokine storm and T-cell lymphopenia in the clinical setting of severe COVID-19. As a computationally predicted, remdesivir-like inhibitor of RNA-dependent RNA polymerase (RdRp)—the central component of the replication/transcription machinery of SARS-CoV-2—silibinin is expected to reduce viral load and impede delayed interferon responses. The dual ability of silibinin to target both the host cytokine storm and the virus replication machinery provides a strong rationale for the clinical testing of silibinin against the COVID-19 global public health emergency. A randomized, open-label, phase II multicentric clinical trial (SIL-COVID19) will evaluate the therapeutic efficacy of silibinin in the prevention of acute respiratory distress syndrome in moderate-to-severe COVID-19-positive onco-hematological patients at the Catalan Institute of Oncology in Catalonia, Spain.

## 1. Introduction

The World Health Organization (WHO) has declared coronavirus disease 2019 (COVID-19) a public health emergency of international concern [1]. The causative agent of the COVID-19 outbreak is the severe acute respiratory syndrome (SARS)-associated coronavirus 2 (SARS-CoV-2), a novel enveloped RNA betacoronavirus [2]. No antiviral drugs are yet available with proven efficacy for SARS-CoV-2 treatment or prophylactic methods to successfully prevent the progression of SARS-CoV-2-driven acute respiratory distress syndrome (ARDS), one of the leading causes of mortality in patients with severe COVID-19. 

From the early clinical experience in China, we learned that the vast majority of COVID-19 patients (>90%) received a diagnosis of pneumonia during hospital admission, followed by ARDS in the more severe cases [3]. A majority of diagnosed patients had peripheral lymphocytopenia, thrombocytopenia, and leukopenia. Patients with severe disease had also elevated levels of serum C-reactive protein and, less commonly, augmented levels of liver transaminases. Laboratory analysis aiming to distinguish severe from mild disease suggested that circulatory inflammatory markers including interleukin (IL)-6, ferritin, and D-Dimer were closely related to severe COVID-19 in adults. In fact, their combined detection had the highest specificity and sensitivity for early prediction of disease severity in patients [4]. These initial findings were consistent with a clinical scenario in which the presence of hypercytokinemia (cytokine storm syndrome (CSS)) and lymphophenia have a main causal role during the transition from first COVID-19 symptoms to viral sepsis and inflammation-induced lung injury, ultimately to pneumonitis, ARDS, respiratory/multiple organ failure, shock, and potentially death. 

In the ensuing months, it has been confirmed that a subgroup of patients with severe disease have CSS that rages 7–10 days after disease onset when the ARDS peaks [5,6]. Immune dysregulation, rather than the level of peak viremia, appears to be one of the major mechanisms through which SARS-CoV-2 infection drives an insufficient (*too-little-too-late*) type I-interferon (IFN) innate immune response, later accompanied by aberrant proinflammatory cytokine secretion from alveolar macrophages that ultimately causes lung damage and reduces survival. Not surprisingly, selective (e.g., IL-6-targeted monoclonal antibodies such as tocilizumab) and non-selective blockade of pro-inflammatory cytokines (e.g., using pan-JAK1/2 inhibitors such as baricitinib) are under evaluation in ongoing clinical trials aimed to reduce the SARS-CoV-2-driven CSS, ameliorate pulmonary inflammation and respiratory distress, and hopefully improve mortality [7,8,9,10,11,12,13,14,15]. 

An optimal therapeutic approach to manage COVID-19 would involve a drug capable of preventing the CSS that dampens adaptive immunity against SARS-CoV-2 while at the same time directly targeting the key molecular machinery driving the virus lifecycle. We here propose that the flavonolignan silibinin—the major bioactive component of the silymarin extract obtained from the seeds of the milk thistle herb (*Silybum marianum*)—[16,17,18] might fulfill such requirements by reducing STAT3-related lung and systemic inflammation in the infected host and directly targeting the RNA replication machinery in the virus. 

## 2. Silibinin: From an Old Remedy to a Direct STAT3 Inhibitor

Originally described as a remedy for the bites of poisonous snakes more than 2000 years ago, the use of silibinin-containing nutraceuticals to treat liver toxicity (e.g., alcoholic and non-alcoholic liver disease), drug-induced liver injury, cirrhosis, mushroom poisoning, and viral hepatitis has been well documented over the last 40 years [19,20]. Since 2013, there has been an ever-growing (molecular) understanding and (clinical) evaluation of the capacity of silibinin to inhibit cell growth of cultured cancer cells and tumor xenografts, to enhance the efficacy of other anti-cancer agents, to reduce the toxicity of cancer treatments, and to prevent and overcome the emergence of cancer drug resistance [21,22,23,24,25,26]. When used with more bioavailable nutraceutical formulations [27], silibinin has proved successful in advanced systemic cancer, a therapeutic activity that was particularly notable in the central nervous system where it provided greater than 4-fold survival benefit in patients with established brain metastases [28]. Importantly, the groundbreaking clinical activity of silibinin was accompanied by low toxicity and reversible secondary effects, and was compatible with the standard-of-care in oncology patients. 

Investigations into the molecular mechanisms involved in the antioxidant, immunomodulatory, antifibrotic, anticancer, and antiviral activities of silibinin have consistently suggested its ability to function as a natural down-modulator of the signal transducer and activator of transcription (STAT3) [29,30,31,32,33,34,35]. Using computational and experimental approaches, we recently delineated the molecular bases of the silibinin-STAT3 interaction. Silibinin synergistically works against STAT3 function by directly blocking the STAT3 Src homology-2 (SH2) domain, which is crucial for both STAT3 phospho-activation and nuclear translocation, and directly targeting also the STAT3 DNA-binding domain (DBD) to prevent the transcriptional activity of STAT3 irrespective of its activation/dimerization status [36]. Silibinin is multi-faceted and impedes the activation, dimerization, nuclear translocation, DNA-binding, and transcriptional activity of STAT3, thereby circumventing both the intrinsic difficulty of efficaciously disrupting protein–protein interactions over a large surface such as those involving SH2-mediated STAT3 dimerization and the previously thought undruggable nature of the STAT3 DBD [36]. Accordingly, cells engineered to overexpress a constitutively active form of STAT3 that dimerizes, binds to DNA and activates transcription spontaneously remain largely unresponsive to the transcriptional and phenotypic effects of silibinin (28, and *unpublished observations*). Importantly, the unique behavior of silibinin as a bimodal SH2- and DBD-STAT3 transcription factor inhibitor in vitro, in situ, and in vivo [28,36] largely explains its ability to fully prevent the hyper-activation of STAT3 imposed by the excessive production of IL-6, a key driver of the dysregulated inflammation in patients suffering from severe COVID-19. 

## 3. Silibinin and STAT3: Targeting the (Reactive) Cytokine Storm

Early studies in lung injury models suggested that STAT3 activation plays a central role during the acute lung inflammatory response to lung tissue damage in a macrophage- and neutrophil-dependent manner [36,37]. Increased alveolar epithelial cell death and phosphorylated STAT3 were common phenotypic traits in patients with ARDS, suggesting the feasibility of targeting STAT3 to modulate pulmonary inflammatory responses. In support of this notion, attenuation of STAT3 signaling suppressed the excessive production of pro-inflammatory cytokines in macrophages and inflammatory cells in several pre-clinical and clinical models, including hyperoxia-, lipopolysaccharide (LPS)-, and sepsis-induced acute lung injury, while additionally promoting lung repair [38,39,40,41,42,43,44,45]. Although scarce, evidence from the phylogenetically-related virus SARS-CoV suggested that STAT3 could operate as a key signaling mediator of the link between virus replication and lung fibrotic responses [46].

Silibinin has proven therapeutic capacity to protect *damaged* tissues by regulating the reactive intensity of cells tasked with establishing a repair program (e.g., macrophages, T-cells, and astrocytes) (Figure 1). Silibinin pre-treatment in mice significantly inhibits LPS-induced recruitment of airway inflammatory cells (macrophages, T-cells, and neutrophils) as well as the production of specific pro-inflammatory cytokines (i.e., IL-1β, TNFα), thereby protecting against lung injury [47,48]. In a mouse model of radiation therapy for lung cancer treatment, which partially mimics the late-phase inflammation and end-stage pulmonary fibrosis related to ARDS in severe COVID-19, silibinin was found to reduce inflammatory cell infiltration in the respiratory tract, to ameliorate inflammation and fibrosis, and to increase survival [49]. The efficacy of silibinin in localized lung tumors was originally found to involve the inhibition of the production and secretion of cytokines from tumor-associated macrophages in a STAT3-related manner [30]. In brain metastases, the suppressive effects of silibinin can be explained in terms of its ability to halt the pro-metastatic program driven by STAT3 in a subpopulation of reactive astrocytes surrounding metastastic lesions [28] (Figure 1). Accordingly, silibinin-inhibited STAT3 signaling in *reactive* (astrocytes) cells to *damaged* (brain metastatic cancer) cells blocks the cytokine secretome of the former to influence immunity responses against metastatic cells, including changes in the activation of CD8^+^ T-cells [28]. 

Early mouse models of SARS-CoV infection predicted that, in patients infected with pathogenic coronaviruses and perhaps other respiratory viruses, the rapid kinetics of viral replication accompanied by delayed type I IFN-I signaling would lead to the pathogenic accumulation of inflammatory monocyte-macrophages (IMMs), resulting in elevated levels of lung cytokines/chemokines (cytokine storm) and impaired virus-specific T-cell responses [50,51]. Such a link between dysregulated inflammatory responses, lung immunopathology, and diminished survival has been confirmed in the present pandemic of COVID-19. Accordingly, the intensity level of the interaction occurring between *damaged* lung epithelial cells and *reactive* IMMs (which switch their phenotype from suppressive/protective to stimulatory/destructive) during the course of SARS-CoV-2-driven systemic inflammation appears to determine, at least in part, the degree of the disease severity in patients [5,6] (Figure 1). In moderate COVID-19 cases, bronchoalveolar macrophage-epithelial interactions promote an increase in IL-6 and a decrease in the counts of total T-cells, particularly CD4^+^ and CD8^+^ T-cells. In severe COVID-19 cases, the macrophage-epithelial interaction promotes a further augmentation of IL-6 (and IL-2R, IL-10, and TNFα), whereas CD4^+^ (including IFNγ-expressing CD4^+^ T cells) and CD8^+^ T-cells markedly decrease in number [5,6]. While the type of macrophage ultimately driving the cytokine storm in severe COVID-19 remains to be unambiguously identified [52], a recently described landscape of lung bronchoalveolar immune cells in COVID-19 using single-cell RNA sequencing revealed that monocyte-derived ficolin 1-positive macrophages, which are highly inflammatory and potent cytokine producers, likewise overwhelm the severely damaged lungs in COVID-19 patients with ARDS [53].

From a mechanistic and therapeutic perspective, the fact that IMMs promote a late and lethal SARS-CoV-2 infection irrespective of the viral load immediately implies that targeted antagonism of such dysregulated response would improve outcomes in patients with severe SARS-CoV-2 infection (Figure 2). Silibinin might be expected to phenotypically integrate the mechanism of action of IL-6-targeted monoclonal antibodies and pan-JAK1/2 inhibitors by directly modulating downstream STAT3 activity in the futile cycle of SARS-CoV-2-*damaged* lung tissues that orchestrate a reactive inflammatory monocyte/macrophage response and sensitize T-cells to apoptosis, resulting in a further dysregulated inflammatory response. Silibinin would thus operate as an immune therapeutic to alleviate the cytokine storm and T-cell lymphopenia in the clinical scenario of a subgroup of severe COVID-19 patients fully meeting ARDS criteria. 

## 4. Silibinin and the RNA-Dependent RNA Polymerase Complex: Targeting Virus Replication 

The IFN-I-dependent immunopathological events, including the recruitment of immunopathogenic immune cells by inflammatory mediators and impairment of T cell responses, which are largely independent of virus replication, are key promoters of SARS-CoV-2 morbidity and mortality. In the delicate balance of virus-host interactions in COVID-19, however, it is also true that it is the extremely rapid and robust replication of SARS-CoV-2—while IFN-I expression is delayed—that initially stimulates lung inflammation, underscoring the relevance of reducing initial viral load through anti-viral interventions (Figure 2). In the latter regard, most of the current therapeutic efforts are directed to pharmacologically target the host-virus interface linking the viral S protein to the angiotensin-converting enzyme 2 receptor in host cells, main 3C-like protease- and papain-like protease-dependent viral replication (a proposed mechanism for lopinavir/ritonavir, ASC09/TMC-310911, or darunavir/cobicistat), and the RdRp/nsp12-driven viral RNA synthesis (a proposed mechanism for remdesivir, favipiravir, emtricitabine/tenofovir alafenamide, ribavirin, or more recently sofosbuvir) [54,55,56,57,58,59,60,61,62,63,64,65,66,67]. Intriguingly, a recent target-based virtual ligand screening study predicted that silibinin might target RdRp/nsp12 [68]. 

RdRp/nsp12 is a central component of a multi-subunit RNA-synthesis complex that additionally requires the assistance of the co-factors nsp7 and nsp8 to ensure the fidelity of faithfully replicating the largest known RNA genome among all the RNA viruses [69,70,71,72,73,74]. The architecture of RdRp/nsp12 appears to be common to all viral polymerases; however, the catalytic mechanism by which SARS-CoV polymerase performs de novo RNA initiation is likely to be distinct from other viruses. Thus, while the nsp12/nsp7/nsp8 complex possesses de novo initiation capacity is dependent on the nsp12 polymerase-active catalytic site, it is noteworthy that the SARS-CoV nsp12 double-stranded RNA exit tunnel has no obstruction to act as a platform for priming nucleotides [74]. Nsp7 and nsp8 heterodimers appear to stabilize nsp12 regions involved in RNA binding while a second nsp8 subunit can play a crucial role in polymerase activity via extension of the template RNA-binding surface. To computationally validate the recent prediction of silibinin targeting the coronaviral RdRp, we took advantage of the 3.1 Å resolution of the SARS-CoV nsp12 polymerase bound to its essential co-factors, nsp7 and nsp8 [74]. 

We employed the SARS-CoV 6NUR crystal structure as a template to generate a homology model of the SARS-CoV-2 RdRp (Figure 3, top). Docking simulations of silibinin over the SARS-CoV-2 nsp12/nsp7/nsp8 model complex produced eleven clusters of docking poses, three of which were predicted to occupy the RNA template tunnel of the coronaviral polymerase. In the absence of the nsp7 and nsp8 cofactors, we similarly predicted the occurrence of three clusters of docking poses occupying the RNA template tunnel (Figure 3, top). Based on these findings, we performed a more detailed comparative docking analysis of silibinin versus currently employed (or predicted) SARS-CoV-2 RdRp-targeted nucleotide analog anti-virals (i.e., remdesivir, ribavirin, and sofosbuvir) in a computational grid centered on the RNA template tunnel (Figure 3, bottom). When the docking results were ranked according to the ascent of the binding energies (up to-9.4 kcal/mol), the silibinin clusters exhibiting the highest affinity were in the low (~100 nmol/L) nanomolar range (Appendix A), which was similar to that predicted for remdesivir (Appendix A) but notably lower than the ranges predicted for ribavirin and sofosbuvir (Appendix A, respectively). 

To verify further these findings, we performed an additional comparative docking analysis of silibinin with remdesivir, ribavirin, and sofosbuvir using the recently reported cryo-electron microscopy analysis of the SARS-CoV-2 full-length nsp12 in complex with cofactors nsp7 and nsp8 at 2.9 Å resolution [75]. The overall architecture of the SARS-CoV-based nsp12-nsp7-nsp8 homology model was similar to that of SARS-CoV-2 with root mean square deviation values of 0.565 (for 6346 atoms; non-reduced form 6M71) and 0.534 (for 6876 atoms; reduced form 7BTF), respectively (Figure 4, top). Noteworthy, the predicted energy bindings of the best clusters of silibinin to SARS-CoV-2 nsp12 (Figure 4, bottom) remained in the submicromolar/low-micromolar range (Appendix A), which was similar or even slightly lower than those predicted for remdesivir and sofosbuvir (Appendix A), and inferior to that in high micromolar range of ribavirin (Appendix A). Nonetheless, silibinin was predicted to interact with well-characterized catalytic residues (e.g., Asp618, Asp623, Asp760, Asp761) as well as with other key residues involved in the nsp12 interaction with the entry path of RNA template/nucleoside triphosphate (e.g., Arg555, Val557, Thr680, Ser682, Asn691) when using either the SARS-CoV-based homology model or the newly characterized SARS-CoV-2 nsp12 structure.

To add protein flexibility and provide additional information about intra- and inter-molecular movements, we performed molecular dynamics (MD) simulations over the course of 100 ns to confirm the kinetic stability of the poses obtained by docking (Appendix A); the root mean square deviation (RMSD) of silibinin versus remdesivir heavy atoms measured after superimposing either nsp12 alone or the nsp12-nsp7-nsp8 hetero-complex on their references structures during the MD simulation were prepared in parallel. From the RMSD simulations, it was easily observed that a majority of the silibinin-containing systems rapidly equilibrated (less than 10 ns) and remained stable until the end of the simulation. In the case of remdesivir, equilibration was obtained later (at around 20 ns) and more fluctuations were visible over the course of simulation. Molecular mechanics Poisson-Boltzmann surface area (MM-PBSA) parameters, which estimate the free energy of the binding of silibinin/remdesivir to SARS-CoV-2 RdRp and are known to show good correlation with the experimentally obtained values, were finally calculated for the entire MD simulation trajectory of 100 ns (or the last 30 ns). MM-PBSA values > 20 kcal/mol and up to 36 kcal/mol were obtained in the case of silibinin, which were in a similar range of those obtained for remdesivir, thereby predicting the strong binding behavior of silibinin to SARS-CoV-2 RdRp. 

The RdRp coronaviral replication/transcription machinery is considered a primary target for new antiviral therapeutics. Therefore, the in silico predicted ability of silibinin to share, at least in part, the ability of nucleotide analogs to occupy the catalytic site of nsp12 [75,76] might support the consideration of silibinin as a new antiviral targeting SARS-CoV-2 RdRp, thereby adding a new therapeutic dimension to the previously recognized ability of silibinin to function as a broad-spectrum antiviral. 

## 5. Silibinin and COVID-19: An Ongoing Clinical Trial

The induction of the JAK/STAT signaling pathway by IFN is known to upregulate a significant number of so-called IFN stimulated genes (ISGs), some of them capable of rapidly kill viruses within infected cells [77]. Therefore, as viral-encoded factors known to antagonize JAK signaling are crucial determinants of virulence, it could be argued that pharmacological blockade of this pathway with available JAK inhibitors (JAKinhibs) or direct STAT3 inhibitors (e.g., silibinin) might produce an impairment of IFN-mediated antiviral response, with a potential facilitating effect on the early-stage evolution of SARS-CoV-2 infection [13,77,78]. Using a JAKinhib to treat a viral infection may play a double role because both type I IFN (IFN-α/β) and type II IFN (IFN-γ) employ the JAK-STAT signaling pathway. Indeed, such a mechanism might explain the increased risk of herpes zoster and simplex infection reported during the development of several JAK inhibitors including baricitinib, upadacitinib, and filgotinib [13,15]. As mentioned above, animal model studies on COVID19-related MERS (Middle East Respiratory Syndrome) and SARS diseases have shown that IFN-α/β therapy might be beneficial in the early inflammatory phase of both diseases, whereas the same therapy can be harmful in the late phase of the diseases [50,51] (Figure 2). 

As nearly 80% of patients with COVID-19 appear to eradicate the SARS-CoV-2 virus via antiviral immune responses (e.g., IFN-α/β), JAK/STAT3 inhibitors may be a better approach when hospital care is needed for COVID-19 patients. Indeed, the peak of SARS-CoV-2 load takes places ~7 days after symptoms onset in those patients with mild disease requiring hospital care. Later, as the viral titer might even decrease in some patients, hyper-inflammation drives the severe phase of the disease, which is accompanied by increased levels of cytokines—all signaling through the JAK signaling pathway. In the scenario of late severe COVID-19, in which the accumulation of IMMs in the lungs as the main source of pro-inflammatory cytokines is associated with fatal disease, targeted antagonism of the JAK pathway with JAKinhibs such as baricitinib (Olumiant^®^)—a dual JAK1/JAK2 inhibitor approved by the European Medicines Agency for moderate-to-severe active rheumatoid arthritis in adults [79,80,81,82]—would improve outcomes given their non-selective capacity to inhibit cytokine release and block cytokine signaling. Using the BenevolentAI’s knowledge graph, a large repository of structured medical information that includes numerous connections extracted from scientific literature by machine learning [83], baricitinib is predicted to interrupt both viral entry and intracellular assembly of viral particles by targeting cyclin G-associated kinase and AP2-associated protein kinase (AAK1), two well-known regulators of endocytosis. Accordingly, baricitinib is being evaluated in clinical trials for the treatment of moderate-to-severe COVID-19 in hospitalized patients either as a single agent (2 mg/day/orally for 10 days; NCT04321993) or combined with antiviral therapy (4 mg/day/orally combined to antiviral therapy ritonavir for 2 weeks, NCT04320277). 

As STAT3 is a latent cytoplasmic transcription factor that can couple with multiple cytokines as an ultimate effector of the JAK/STAT pathway, an important question that remains unexplored is whether STAT3 might be a better target than its upstream JAK activators to clinically manage COVID-19. We recently reasoned that the systemic manipulation of STAT3 signaling with silibinin and its putative, direct anti-viral activity can be trialed as a safe therapy to reduce inflammation and viral replication in a population of SARS-CoV-2-infected oncology patients. The Spanish Agency for Medicine and Health Products (*Agencia Española de Medicamentos y Productos Sanitarios*; AEMPS)—the regulatory agency that oversees the quality, safety, and efficacy of pharmaceutical and medical devices in Spain—has prioritized the approval of a randomized (1:1), open-label, phase II multicentric clinical trial to evaluate the efficacy of silibinin supplementation in the prevention of ARDS in moderate-to-severe COVID-19-positive onco-hematological patients undergoing systemic treatment or having completed treatment <1 year ago (Figure 5). 

The so-called SIL-COVID19 trial will be conducted in two phases: a non-randomized safety phase and a randomized phase. During the clinical study design, we acknowledged that the achievement of a bona fide, clinically relevant activity of silibinin remains controversial in human trials. Then, we decided to take advantage of the so-called Eurosil^85^ formulation, a patent extraction process that augments the oral absorption and bioavailability of silibinin, exhibiting the highest permeability rate in models of human intestinal absorption, that has been shown to exhibit significant clinical activity in cancer patients with advanced systemic disease [24,27,28]. Although milk thistle extracts such as Eurosil^85^ are known to be well-tolerated, with minimal toxic or adverse effects being observed in clinical trials, and co-administration of silibinin with multiple antiretrovirals (e.g., darunavir-ritonavir in HIV-infected patients) has been shown to be safe without a need for dose adjustment of anti-viral therapy [86], the SIL-COVID19 trial will evaluate oral bioavailability, single and multidose pharmacokinetics, and safety of the Eurosil^85^ formulation in a lead-in safety cohort in which to test whether silibinin supplementation combined with best supportive care according to physician’s choice exceeds a safety threshold, and will provide the phase II-recommended dose. The phase II will then assess the impact of silibinin on the severity and progression of COVID-19 when combined with best supportive care according to physician’s choice. The primary endpoint of the study will be efficacy in terms of the clinical status at a given day using a scale of death, need for mechanical ventilation or intensive care unit (ICU) admission, non-invasive ventilation or high-flow oxygen devices, low-flow supplemental oxygen, not requiring supplemental oxygen/no longer requiring ongoing medical care, and not hospitalized. Clinical responses in the experimental and control arms will be compared for statistical difference. Secondary outcome measures will include toxicity/tolerability profiles as well as circulating markers of systemic cytokine release syndrome, vascular permeability and leakage, thrombosis, pulmonary embolism, and coagulopathy with the aim of uncover early biomarkers of poor evolution or response of COVID-19 patients to silibinin-based treatment. 

## 6. Silibinin and SARS-CoV-2: An Early-Intervention in Older Individuals at Risk of Severe COVID-19? 

The infection rates, severity, and lethality of SARS-CoV-2/COVID-19 are substantially higher in the population aged 60 and older. Despite the limited data available, statistics from the COVID-19 pandemic strongly support the notion that SARS-CoV-2 is a gerolavic (from Greek, *géros* “old man,” and *epilavís*, “harmful”) virus that disproportionately affects the elderly [87]. Animal models of SARS-CoV pathogenesis corroborating the age-related susceptibility to COVID-19 disease have demonstrated that the age-driven remodeling of the immune response (i.e., immunosenescence) could play a central role in the enhanced susceptibility of the elderly to the severity and lethality of SARS-CoV-2 infection [88,89]. In addition to immunosenescence, other cellular senescence phenomena in lung-epithelial and -mesenchymal cells might promote chronic airway remodeling and sustain chronic airway inflammation through the well-known capacity of senescent cells to produce large amounts of inflammatory cytokines (e.g., IL-6) as a result of the senescence-associated secretory phenotype (SASP) [90,91]. As the SASP could augment cellular senescence in surrounding cells, such a vicious feedback loop certainly generates a pathogenic condition to infections with viruses that can cause a very brisk cytokine reaction such as SARS-CoV-2. Importantly, however, if an exaggerated lung-associated cell senescence is a key co-morbidity impacting on the age-dependent prognosis of COVID-19 patients [92], it then follows that breaking this chronic cycle may help restrain the risk of COVID-19 aggressiveness and lethality in elderly individuals (Figure 5). It is therefore plausible that the immunomodulatory and senoremediative actions of silibinin could go beyond treatment and may provide a preventative measure before elderly individuals are exposed to SARS-CoV-2. Extensive JAK/STAT pathway studies in aging have demonstrated that this pathway plays a major role in regulating cytokine production as part of the SASP [93] and that STAT3 activation is relevant to the lung-cell senescence program [94,95,96]. Although there is currently no evidence for a senolytic activity of silibinin [97], it should be noted that other silibinin-related plant flavonoids such as quercetin have proven beneficial at reducing senescent cell burden and chronic sterile inflammation [98]; a systematic review of the efficacy of dietary flavonoids on upper respiratory tract infections and immune function in healthy adults have suggested their prophylactic value with no apparent adverse effects [99]. 

It might be argued that because STAT3 appears to direct seemingly contradictory pro- and antiviral responses during the acute phase response of viral infection [100,101], STAT3 inhibition might be exploited by SARS-CoV-2 to evade IFN-related host innate immunity [102]. Intriguingly, however, it has recently been discovered that ACE2, the entry receptor of SARS-CoV-2, is a previously unrecognized human ISG in airway epithelial cells, thereby suggesting that SARS-CoV-2 might exploit species-specific IFN/STAT-driven upregulation of ACE2—a tissue-protective mediator during lung injury—to enhance infection [78]. Moreover, SARS-CoV-2 infection via ACE2 can activate STAT3, which in turn can activate the so-called IL-6 amplifier [103], a mechanism for the hyperactivation of STAT3 that might trigger a positive feedback capable of self-amplifying the IL-6/STAT3 signaling in lung alveolar epithelial cells, thereby accelerating the transition from virus replication/spreading to COVID-19 clinical stages. In this scenario, it might therefore be possible to conduct clinical trials on the protective activity of silibinin for therapeutic development during the initial phases of SARS-CoV-2 infection in residential care nursing homes and elderly day-care centers where old, frail individuals exhibit an elevated pre-disposition for delayed IFN-I signaling (Figure 5), leading to dysregulated IMM response, lung immunopathology, and lethal pneumonia. 

## 7. Silibinin Against the Lifecycle of SARS-CoV-2: Beyond Host STAT3 and Viral Replication

Although generally overlooked or underestimated, there is strong clinical evidence on the safety and efficacy of intravenous and oral silibinin to induce potent antiviral responses capable of rescuing patients chronically infected with hepatitis C virus (HCV)—an enveloped RNA virus belonging to the Flaviviridae family—that fail to respond to combinations of IFN and anti-virals such as ribavirin [104,105,106,107,108,109,110,111]. The Food and Drug Administration (FDA)-approved drug sofosbuvir has recently been proposed as an antiviral for the SARS-CoV-2 based on the similarity of the replication mechanisms of HCV and coronaviruses [67,75]. Following this reasoning, the capacity of silibinin to reduce viral loads in non-IFN responders along with its recognized ability to exert direct anti-viral effects against flaviviruses (HCV and dengue virus), togaviruses (Chikungunya virus and Mayaro virus), influenza virus, human immunodeficiency virus, and HBV [112], might be considered to develop prophylactic interventions in individuals facing significant risk of developing COVID-19 (e.g., close contacts, households, and healthcare workers). In the latter regard, it should be noted that the anti-COVID-19/SARS-CoV-2 activity of silibinin might involve additional (STAT3- and viral RNA polymerase-independent) mechanisms. For example, clathrin-dependent cellular trafficking is a key mechanism for SARS-CoV-2 entry into host cells. Accordingly, the FDA-approved drug chlorpromazine, an inhibitor for clathrin-dependent endocytosis, has been found to exert inhibitory effects on the entry of coronaviruses including MERS-CoV and SARS-CoV [113]. Of note, silibinin has been shown to potently hinder the entry of HCV, reovirus, vesicular stomatitis, and influenza viruses by slowing-down trafficking through clathrin-coated pits and vesicles [114] (Figure 6). 

The capacity to inhibit early steps of viral infection by widely affecting endosomal trafficking of virions supports silibinin as a potential broad-spectrum antiviral therapy. In another example, inhibition of endoplasmic reticulum (ER)-resident α-glucosidases, which sequentially trim the terminal glucose moieties on the N-linked glycans attached to nascent glycoproteins [115], has been shown to impair the S protein-mediated entry of SARS-CoV by altering the glycan processing of ACE2 and, consequently, post-receptor binding mechanisms [116]. Thus, suppression of ER α-glucosidases not only can disrupt the morphogenesis of a broad spectrum of enveloped viruses including SARS-CoV, but also impede their entry by altering the glycan structures of their cellular receptors. Indeed, ER α-glucosidase inhibitors have been proposed as host function-targeting broad-spectrum antiviral agents that might be particularly attractive for treatment of respiratory tract viral infections (e.g., COVID-19) caused by enveloped RNA viruses (e.g., SARS-CoV-2), with a short window for medical intervention [117,118]. Although preliminary, when evaluating the mechanistic insights underlying the immune-sensitizing effects of silibinin, we recently observed its capacity to modulate the N-linked glycan decoration of nascent immunosuppressive molecules at the ER of tumor cells (unpublished observations). Moreover, such N-linked glycosylation-targeted activity of silibinin was compatible with in silico models predicting its capacity to directly target the catalytic site of human ER α-glucosidase (*unpublished observations*). 

## 8. Conclusions

The 2019 SARS-CoV-2 coronavirus outbreak is causing a global pandemic with hundreds of thousands of infections and deaths worldwide. Whereas numerous pharma and biotech companies and academic institutions are racing to develop vaccine candidates for effective COVID-19 prevention, the rapid global spread of COVID-19 has stressed the need for new therapeutics. Currently, three broad groups of anti-coronavirus treatments are being investigated for the prevention and treatment of the life-threatening SARS-CoV-2/COVID-19, namely: (1) those aimed at dampening the exaggerated host immune response; (2) those aimed at blocking viral replication and survival in host cells, and (3) those aimed at halting viral entry into host cells. Here, we present a comprehensive review of the evidence-based research into the multi-faceted capacity of silibinin to target the host cytokine storm and the virus lifecycle to clinically manage COVID-19/SARS-CoV-2 infection (Figure 6). We acknowledge that measurement of the therapeutic efficacy requires randomized trials of silibinin therapy. Accordingly, a clinico-translational research experience will be conducted at the Catalan Institute of Oncology in Catalonia (Spain) for testing the therapeutic efficacy of silibinin in oncology patients hospitalized with COVID-19. Nevertheless, our present work aims to provide a basis for the design of new silibinin-based antiviral therapeutics or supportive care approaches against the COVID-19, a global public health emergency. 

## Figures and Tables

**Figure 1 jcm-09-01770-f001:**
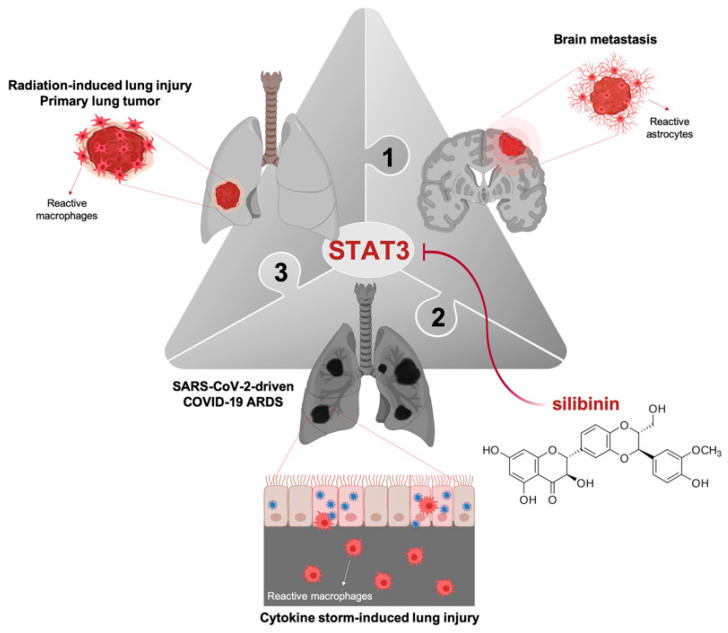
Silibinin as a direct STAT3 inhibitor: regulating the response intensity of reactive reparative cells to *damaged* tissues. STAT3 is a master checkpoint regulator of the interface between cytokines, inflammation, and immune response against various types of tissue damage including viral infections. By operating as a direct inhibitor of STAT3, silibinin regulates the response intensity of reparative cells (e.g., macrophages, astrocytes) to damaged tissues (e.g., radiation-induced lung injury, primary lung tumor, brain metastasis). In the ongoing scenario of the 2019—2020 SARS-CoV-2 pandemic, in which the futile accumulation of inflammatory macrophages in lungs as the main source of pro-inflammatory cytokines associates with fatal disease in a subgroup of severe COVID-19 patients fully meeting acute respiratory distress syndrome (ARDS) criteria, targeted antagonism of STAT3 with silibinin might ameliorate COVID-19 morbidity and mortality.

**Figure 2 jcm-09-01770-f002:**
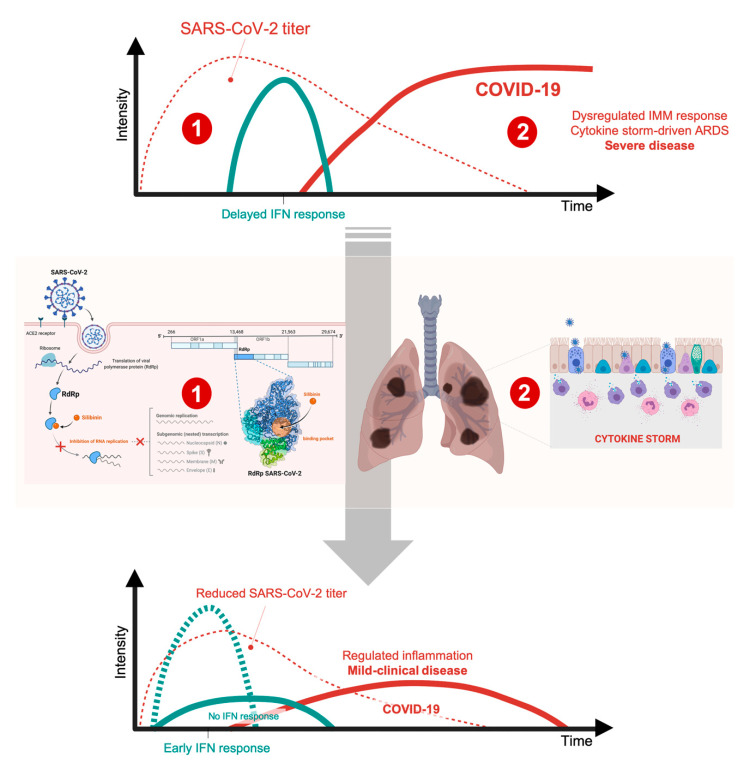
Silibinin: a putative regulator of the *too-little-too-late* interferon response in COVID-19. Whereas it is well known that type I-interferon (IFN)-induced anti-viral response is among the earliest and most potent of the innate responses to fight viral infection, the timing of IFN-I response relative to virus replication might be key for SARS-CoV-2 infection outcome. (1) The extremely rapid and robust replication of SARS-CoV2—while IFN-I expression is delayed—is one of the initial triggers of lung inflammation, therefore highlighting the relevance of reducing initial viral load through direct anti-viral interventions. By targeting the central component of the replication/transcription machinery of SARS-CoV-2, namely the viral polymerase RdRp, silibinin is expected to reduce viral load and/or impede delayed interferon responses. The early antagonism of several SARS-CoV-2 proteins to the IFN response delays or prevents the innate immune response. Delayed IFN signaling, however, further orchestrates inflammatory monocyte/macrophage (IMM) responses and sensitize T-cells to apoptosis, which results in a further dysregulated inflammatory response, cytokine-driven acute respiratory distress syndrome (ARDS), and severe/fatal disease in a subgroup of COVID-19 patients. (2) Silibinin might be expected to phenotypically integrate the mechanisms of action of IL-6-targeted monoclonal antibodies and pan-JAK1/2 inhibitors to alleviate the pathological accumulation of inflammatory macrophages as the source of the cytokine storm and T-cell lymphopenia associated with fatal COVID-19 disease. We acknowledge that the proposed viral RdRp-targeted (1) and host STAT3-targeted (2) mechanisms of silibinin are highly intertwined. On the one hand, the (pro-/anti-) responses of STAT3 signaling to virus infection are complex both in a virus-specific and in a stage-specific manner in the virus lifecycle. On the other hand, the host IFN response could promote the ability of SARS-CoV-2 to maintain cellular targets in neighboring human upper airway epithelial cells via up-regulation of the SARS-CoV-2 receptor ACE2 in a STAT1-related manner. As the competition phenomena and distinct dynamics of the STAT3/STAT1 duo in pro- (e.g., IFN-I) and anti-inflammatory pathways might constitute an evolutionary-conserved mechanism of the host to tightly control the immune response while avoiding tissue damage, further studies should clarify whether STAT3 participates in the net protective or detrimental role of type I IFN depending on the stage of SARS-CoV-2 infection. In animal models, IFN-I administration shortly after SARS-CoV infection has positive effects and protects mice from lethal infection, whereas later administration IFN-I fails to inhibit viral replication and has side-effects (i.e., severe COVID-19-like increased infiltration and activation of monocytes/macrophages/neutrophils in the lungs accompanied by enhanced pro-inflammatory cytokine expression) that result in fatal pneumonia from an otherwise sublethal infection. Accordingly, although promising findings have been observed upon IFN-I treatment or IFN-based combination therapy with lopinavir/ritonavir, ribavirin, or remdesivir in pre-clinical animal models, mixed results have been found in humans. Understanding if STAT3 is one of the host restriction/promoter factors targeting SARS-CoV-2 lifecycle in a IFN-I/ACE2-dependent/independent manner may provide better strategies to dissociate the dual roles of IFN-I in SARS-CoV-2 infection.

**Figure 3 jcm-09-01770-f003:**
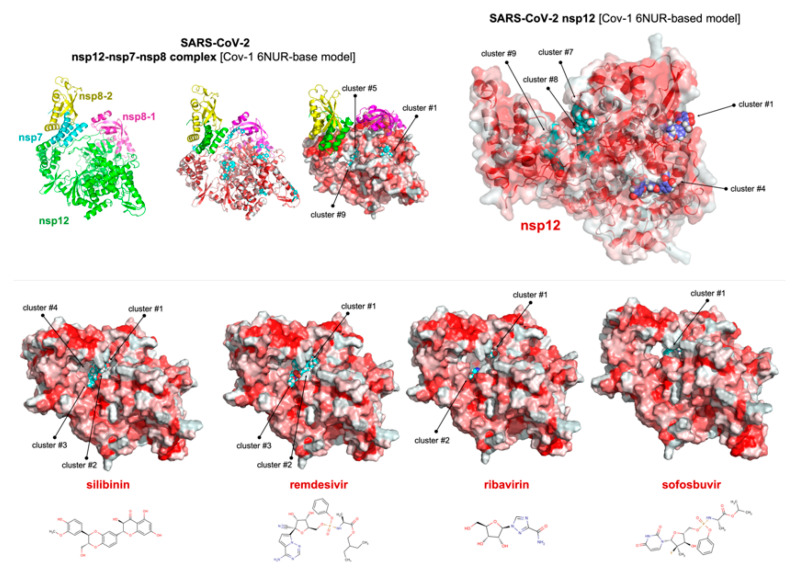
Silibinin is predicted to target SARS-CoV-2 viral polymerase RdRp (I). Representations of a SARS-CoV-1 6NUR-based homology model of the SARS-CoV-2 viral polymerase RdRp showing the computationally predicted locations of silibinin, remdesivir, ribavirin, and sofosbuvir clusters in the nsp12 catalytic subunit of the nsp12-nsp7-nsp8 complex. The protein has been represented as a function of the hydrophobicity of its surface amino acids and the Na^+^ and Cl^-^ ions have been eliminated to facilitate visualization. Figures were prepared using PyMol 2.3 software.

**Figure 4 jcm-09-01770-f004:**
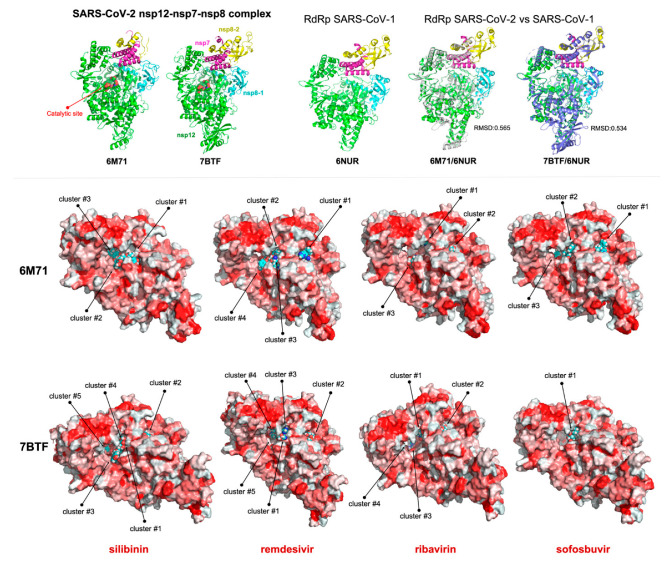
Silibinin is predicted to target SARS-CoV-2 viral polymerase RdRp (II). Representations of the SARS-CoV-2 viral polymerase RdRp in non-reduced (6M71) and reduced (7BTF) conditions showing the computationally predicted locations of silibinin, remdesivir, ribavirin, and sofosbuvir clusters in the nsp12 catalytic subunit of the nsp12-nsp7-nsp8 complex. The protein has been represented as a function of the hydrophobicity of its surface amino acids and the Na^+^ and Cl^-^ ions have been eliminated to facilitate visualization. Figures were prepared using PyMol 2.3 software.

**Figure 5 jcm-09-01770-f005:**
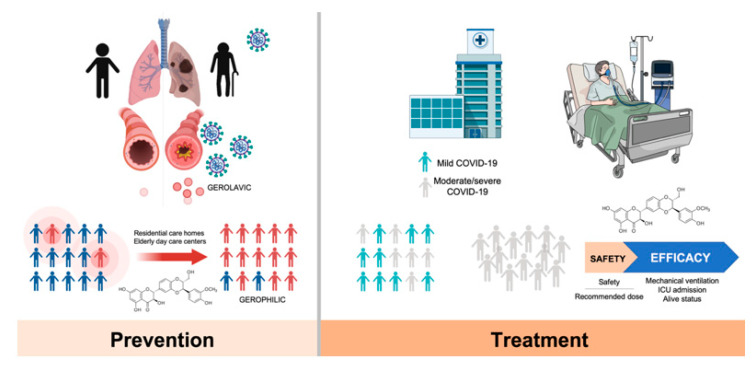
Silibinin in the prevention and treatment of COVID-19. *COVID-19* prevention. Severe COVID-19 illness and death is largely more common in the population aged 60 and older. The gerophilic and gerolavic traits of SARS-CoV-2 infection might rely on immunosenescence and other lung-associated cellular senescence phenomena promoting chronic airway remodeling and sustain chronic airway inflammation. The immunomodulatory and senoremediative activities of silibinin might ameliorate the co-morbid features of aging lungs (e.g., chronic inflammation, airway remodeling, function decline) thereby reducing the risk of COVID-19 aggressiveness and lethality in elderly individuals. In frail, elderly individuals with an elevated pre-disposition to experience delayed IFN-I signaling leading to dysregulated inflammatory monocyte/macrophage responses, lung immunopathology, and lethal pneumonia, the direct antiviral actions of silibinin might also be explored as a prophylactic intervention. *COVID-19 treatment.* An ongoing clinico-translational research study is being conducted at the Catalan Institute of Oncology in Catalonia (Spain) for testing the therapeutic efficacy of silibinin in oncology patients hospitalized with moderate/severe COVID-19. A Jung’s two-stage design for randomized phase II trials with a prospective control was used to estimate the sample size [84]. To keep the sample size small and the study period short, we employed a relatively large type I error (α = 15%) and short-term outcome variable (i.e., the number of responses as primary endpoint), which will allow for early termination of the study if the silibinin-containing arm fails to show efficacy at the interim analysis. The combination of silibinin plus best supportive care will be considered worthwhile if there is a response rate of 90% in the experimental arm (versus the expected 70% in the control arm [85]). By setting an α level of 0.15, a power of 0.80, a balanced allocation (1:1), and an expected drop-out rate of 10%, the sample size will be 16 patients for both arms to ensure the assessment of response rates in 14 patients in each arm at the first stage. Only if at least two more patients achieve the desirable clinical response in the silibinin-containing arm than in the reference arm, and provided no safety issues are identified, would the clinical trial proceed to the second stage. In such a case, an additional recruitment of 25 patients for both arms (to ensure the assessment of response rates in 22 patients in each arm), will proceed. The silibinin-containing arm will be considered effective if four or more additional patients achieve a clinical response in comparison with the reference arm at the end of the study (*n* = 82 patients at the planned final sample size).

**Figure 6 jcm-09-01770-f006:**
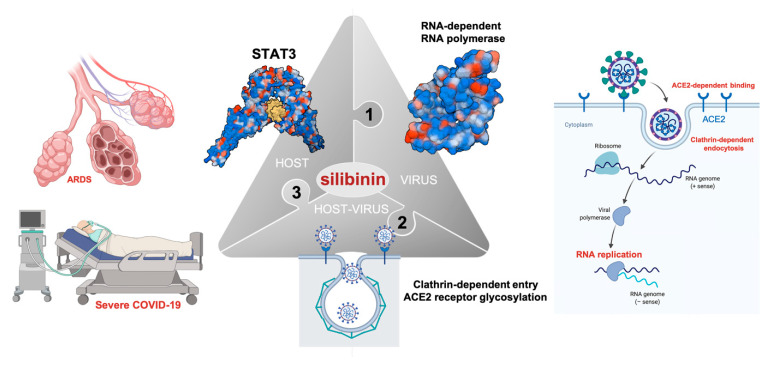
Silibinin: A multi-faceted approach to SARS-CoV-2/COVID-19. The SARS-CoV-2/COVID-19 pandemic is a current threat to human health. As the application of antiviral drugs is expected to provide an immediate and direct control of SARS-CoV-2 infection, a major strategy for managing COVID-19 patients might involve targeting conserved viral proteins critical for viral replication such as the viral polymerase RdRp. A second approach might involve the targeting of key virus-host interactions such as the ACE2-dependent binding and clathrin-dependent endocytosis processes enabling the entry of SARS-CoV-2 into the host cell. Finally, the targeting of host cellular mechanisms that are triggered to defend against SARS-CoV-2 infection (e.g., STAT3-driven reactive immune-inflammation) might impede not only the viral exploitation of cellular responses (e.g., host type I interferon) in support of its efficient replication but, perhaps more importantly, the overproduction of pro-inflammatory cytokines and the overactivation of immune cells that ultimately drives an acute respiratory distress syndrome (ARDS) as one of the leading causes of mortality in a subgroup of patients with severe COVID-19. The multi-faceted ability of silibinin to target all the therapeutically relevant clinico-molecular traits of SARS-CoV-2 infection provides a strong rationale for the clinical testing of silibinin against the COVID-19 global public health emergency.

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
