# Peer review of "Silibinin and SARS-CoV-2: Dual Targeting of Host Cytokine Storm and Virus Replication Machinery for Clinical Management of COVID-19 Patients"

_jcm, 2020, doi:10.3390/jcm9061770_

Round 1
Reviewer 1 Report
The manuscript titled " Silibinin and SARS-CoV-2: Dual targeting of host cytokine storm and virus replication machinery for clinical management of COVID-19 patients" is a perspective based article discussing the possible use of Silibinin in the antiviral therapy of COVID-19. The manuscript itself presents the reader with excellent possible therapeutic and functional mechanisms of the aforementioned drug. The authors have also explained the possible mechanisms through well explanatory diagrams. There are only few questions which arise while reading the manuscript:
How do the authors feel about possible side-effects arising from the antiviral therapy in various age groups or pregnant women specially?
Some of the previous therapies in history have failed due to several reasons including antibody dependent enhancement or suboptimal dosing. What do the authors feel these can be prevented in case of the clinical trials opted for Silibinin?
Reviewer 2 Report
In this article, the authors propose the use of the flavonoligan Silibin for clinical management of COVID-19 patients. Indeed, using computational and experimental approaches, the authors have initially reported that Silibinin is a direct inhibitor of the transcription factor STAT3 (PMID: 29660364) thereby preventing STAT3 activation by pro-inflammatory cytokines such as IL-6. Hence as a direct STAT3 inhibitor, Silibinin could be use to regulate the response intensity of reactive cells to damaged tissues for instance lung injury following SARS-CoV-2 infection (SARS-Cov-2-induced COVID-19 acute respiratory distress syndrom or ARDS). Silibinin can also be expected to phenotypically integrate the mechanism of action of neutralizing IL-6 antibodies and pan-JAK1/2 inibitors by directly moudulating dwnstream STAT3 activity in the context of SARS-CoV-2-induced tissue damages that are induced by the inflammatory monocyte/macrophage response.
Moreover, based on the structure of the SARS-CoV RdRp/nsp12 polymerase (RNA-dependent RNA polymerase), the author used this template to generate an homology model of the SARS-Cov-2 RdRp. Silibinin is then predicted to target the SARS-Cov-2 viral polymerase RdRp with an affinity similar to that predicted for remdesivir but lower than ribavirin and sofosbuvir. Therefore, Silibinin appears as a compound that can be used both in the prevention and treatment of COVID-19 since it may have a mutli-faceted approach against SARS-CoV-2 infection. Based on the context of the current SARS-CoV-2 outbreak, this manuscript is obviously relevant and interesting. While Silibinin has been proposed to have a protective effect against other virus as HIV, HCV or HBV, a putative role of this drug against coronavirus has never been been proposed. The conclusions raised by the authors are well supported by the presented arguments and evidence which fit well with the addressed questions. Finally The paper is well written and the text is clear as well as easy to read. My only concern is that their conclusions come from in silico analyses so that it remains to be determined in patients whether Silibinin is really efficient, as suggested by the authors.
Reviewer 3 Report
The content of the paper is highly suggestive as a drug that contributes to the control of pneumonia in COVID-19, and it is very interesting to introduce the suppression of proliferation of SARS-CoV and the therapeutic application to COVID-19.
The comments for each paragraph are listed below.
- Regarding the basis for focusing on silibinin, if there is a feature that is different from other anti-inflammatory agents such as transcription factor inhibitors, it may be better to include it.
- It has been described in the first paragraph that silibinin has been used for a long time. However, for the purpose of this paper, it is easier to read by focusing on inflammation control related to viral infection.
- In SARS-CoV infection, since the IFN response based on delayed I IFN-I signaling is reduced, it is expected that the JAK-STAT system signal is reduced. Please summarize and describe the basis and mechanism of action that silibinin suggests direct binding to STAT3.
- Regarding the point that silibinin acts on RdRp, it is not necessary to introduce all virus-derived molecules introduced in the first half. It is easier to understand by focusing on silibinin and related nsp12, nsp7 and nsp8, and then describing the relationship with RdRp. It is expected that the virus growth cannot be completely controlled, but is it dependent on the ability to bind to RdRp or the intracellular concentration of silibinin?
- It is hard to understand what the Line 330 means. Please add some more suggestive expressions. Line 334-350, please summarize the contents briefly. L354: Is STAT3 activation higher in tumor patients than in non-tumor patients infected with COVID-19?
- Elderly people become more severe when infected, but is there no risk that silibinin administration will further lower the innate immune function? Is there any information about the risk of co-administration with other therapies?
- Regarding the pharmacological action in the body, the dose and effective dose, half-life, stability, and transportation of medicinal components are important, but what method is effective for COVID-19 from existing biopsy? Or?
Please consider revising your paper.
